# Periodic Multilayer for X-ray Spectroscopy in the Li K Range

Vladimir Polkonikov [1], Nikolai Chkhalo [1], Roman Pleshkov [1], Angelo Giglia [2,*], Nicolas Rividi [3], Emmanuelle Brackx [4], Karine Le Guen [5], Iyas Ismail [5] and Philippe Jonnard [5]

1. Institute of Physics of Microstructure, Russian Academy of Sciences, 603087 Nizhny Novgorod, Russia; polkovnikov@ipmras.ru (V.P.); chkalo@ipmras.ru (N.C.); Pleshkov@ipmras.ru (R.P.)
2. Instituto Officina Dei Materiali—Consiglio Nazionale Delle Ricerche, 34149 Trieste, Italy
3. Service Camparis, UMR7154—CNRS, Institut de Physique du Globe, CEDEX 05, 75252 Paris, France; nicolas.rividi@sorbonne-universite.fr
4. Commissariat à L'énergie Atomique et aux Énergies Alternatives, Direction des Énergies, Institut Pour les Sciences et Technologies Pour une Economie Circulaire des Energies bas Carbone, Département de Recherche sur les Procédés Pour la Mine et le Recyclage du Combustible, Univ Montpellier, Marcoule, 34095 Montpellier, France; emmanuelle.brackx@cea.fr
5. Laboratoire de Chimie Physique—Matière et Rayonnement, Faculté des Sciences et Ingénierie, Sorbonne Université, Unité Mixte de Recherche du Centre National de la Recherche Scientifique, 4 Place Jussieu, CEDEX 05, 75252 Paris, France; karine.le_guen@sorbonne-universite.fr (K.L.G.); iyas.ismail@sorbonne-universite.fr (I.I.); philippe.jonnard@sorbonne-universite.fr (P.J.)
* Correspondence: giglia@iom.cnr.it; Tel.: +39-040-375-8422

**Featured Application: The designed periodic multilayer, implemented on electron microprobe analyzers or scanning electron microscopes, will improve wavelength dispersive X-ray spectroscopic investigations in the range of the Li K emission.**

**Abstract:** X-ray spectroscopy of lithium is very difficult, even impossible, with wavelength dispersive spectrometers commonly deployed on scanning electron microscopes or electron microprobe analyzers. This is due to the absence of crystals and lack of efficient periodic multilayer for this spectral range, around 50 eV. To address this issue, we propose using a Be/Si/Al multilayer having a period of about 29 nm. The multilayer was deposited by magnetron sputtering and its reflectivity measured as a function of the glancing angle in the spectral range of the Li K emission and as a function of the incident energy up to ~200 eV. This characterization demonstrates that the designed multilayer is suitable to efficiently perform spectroscopy in the range of the Li K emission in terms of reflectance (0.32 at 51.5 eV), bandwidth (around 3.5 eV) and rejection of high order diffracted radiation.

**Keywords:** nanoscale multilayer; X-ray emission spectroscopy; lithium; multilayer mirror; beryllium

## 1. Introduction

Most chemical elements in inorganic samples can be quantitatively analyzed by X-ray microanalysis. This nondestructive technique, classically implemented on scanning electron microscopes (SEM) and electron probe microanalyzers (EPMA), is used to quantify (determine the weight fraction of) elements present in solid samples from the measurement of the characteristic X-ray intensities emitted by these elements under electron irradiation. However, lithium, despite its many applications, notably in the energy domain [1], is a light element that cannot currently be measured with this standard X-ray analytical technique. Today, lithium analysis thus requires the use of either synchrotron radiation facilities, not readily accessible, or destructive techniques such as laser ablation inductively coupled plasma mass spectrometry (LA-ICP-MS) [2] or laser induced breakdown spectroscopy (LIBS) [3]. Moreover, LIBS analysis is only semiquantitative due to major matrix effects, and LA-ICP-MS analysis requires analytical spots often larger than the size of many structures in materials, which are of micrometric size (batteries, ceramics, minerals).

The Kα spectrum of lithium (in the 45–55 eV spectral range [4]) has been studied for a long time with high spectral resolution grating spectrometers [5–9]. However, there has been recent progress in equipping laboratory instruments with spectrometers dedicated to the ultrasoft X-ray range and particularly for the analysis of the Li K emission [10,11]. To disperse the X-ray radiation, they rely on gratings or zone plates and thus should be used as accessories on SEM and EPMA, which come in addition to the standard bent crystal spectrometers. A much simpler and cheaper way would consist of using a periodic multilayer as a dispersive element, which can be set up in the crystal spectrometers of SEM or EPMA. In this case, the user can continue to work with his standard procedure to obtain the Li K spectrum necessary to perform quantification.

Owing to the long wavelength of the Li K radiation (around 25 nm), there is no available crystal to perform spectroscopy, but nanoscale periodic multilayers, whose period can be suited to the desired spectral range, should be used, as this is commonly done to analyze low-Z elements [12–17]. In order to perform a reliable and precise quantification, it is necessary to obtain the most intense possible spectrum, free of spectral interferences. This translates into the following requirements for the design of the multilayer, which should:

- present a high reflectance in the Li K range;
- present a narrow bandwidth;
- minimize the contribution of the high diffraction orders;
- have a period leading to a Bragg angle at the center of the angular range spanned by standard spectrometers used in SEM and EPMA.

In this context, Be-based periodic multilayers, i.e., multilayers in which one of the layers is a Be thin film, are of high interest [18–20]. In these works, the Be/Si/Al structures were studied for the 17–30 nm (40–75 eV) wavelength range. The combination of only weakly absorbing materials in the period allows one to increase the extinction depth in comparison with the case of materials with high absorption: for example, Mo/Si or $B_4C$/Si [21]. This increases spectral selectivity (decreases bandwidth). The reflection coefficient remains high due to the fact that a larger number of periods of the structure diffract when the radiation is reflected than in the case of using strongly absorbing materials.

It is important to note the role of the silicon layer as an element smoothing out the roughness. Theoretically, a two-component Be/Al structure should have a high reflection. However, it was shown in [18–20] that such a structure has a high (up to 2.3 nm) interfacial roughness. The introduction of a thin (1–2 nm) silicon layer reduces the roughness by a factor of 1.5 to 2. The advantage of using a combination of three elements, aluminum, silicon and beryllium in our case, is associated with the suppression of the second and third orders of diffraction lying behind the corresponding absorption edges of Al, Si and Be.

This article examines the reflectivity of Be/Si/Al multilayers optimized for use as dispersing elements in crystal spectrometers equipping SEM and EPMA.

## 2. Materials and Methods

Theoretical simulations of optical properties of multilayer stacks show that the structure Be (12.5 nm)/Si (2 nm)/Al (14.5 nm) has the highest reflection in the 47–57 eV range, at grazing angles around 30°. This structure, where the order of the layers starts from the surface, was the aim. The penetration depth is quite small in this photon energy range, and simulations show that only a few layers are necessary, 30 periods (90 layers), to reach the maximum of reflectance. The reflectance is 99.7% of this maximum value for a stack with 20 periods (60 layers) and 93% for a stack with 10 layers (30 layers). Thus, the Be/Si/Al period was reproduced 20 times.

The stack was deposited by *dc* magnetron sputtering on a smooth (0.1–0.2 nm) silicon substrate. Deposition system and the technological process for the synthesis of beryllium-containing structures are described in detail in [22]. In this experiment, high-purity (99.998%) argon at a pressure of ~0.1 Pa was used during the sputtering process, while the base pressure was lower than $10^{-4}$ Pa. The electric power on the magnetrons was

112 W for Be, 152 W for Al and 162 W for Si. In these conditions, the growth rate of the films was 0.06 nm/s for Be, 0.21 nm/s for Al and 0.16 nm/s for Si.

Following deposition, the hard X-ray reflectivity of the sample was measured by means of a PANalytical X'PertPro four-crystal high-resolution diffractometer at the wavelength of 0.154 nm (Cu Kα radiation). The basic parameters of the structure, such as period, film thickness and interfacial roughness, were extracted from the fit of the experimental reflection curve, obtained in the θ–2θ mode, using the Multifitting software for reflectometric reconstruction of multilayer structures [23]. Figure 1 shows the experimental angular dependence of the reflection coefficient and the result of the fit.

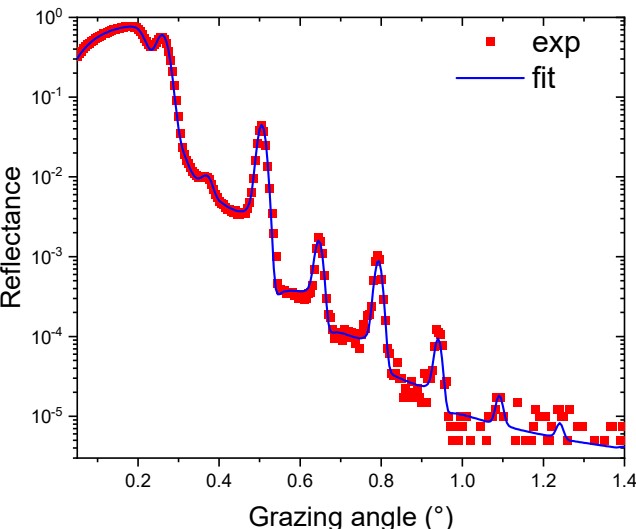

**Figure 1.** Angular dependence of the reflectance at the wavelength of 0.154 nm: experiment, points; fit, solid line.

The parameters of the synthesized structure are determined as Be (12.6 nm–1.4 nm)/Si (2.0 nm–0.8 nm)/Al (14.4 nm–1.6 nm), where the first number in parenthesis is the thickness of the layer and the second number its interfacial roughness. The uncertainty of the thickness and roughness is estimated to be 0.1 nm. The deposited thicknesses are closed to the intended ones and result in a period of the stack equal to 29 nm.

X-ray reflectivity measurements in the Li K spectral range were performed on the BEAR beamline [24] at the Elettra synchrotron radiation facility. Incoming and reflected intensities were measured with a SXUV 100 photodiode with an integration time of 0.5 s. First, measurements as a function of the glancing angle in θ–2θ mode were performed for six incident photon energies spanning the spectral range of the Li K emission. Monochromatic incident radiation was *s*-polarized. Second, measurements were performed as a function of the incident photon energy at given angles corresponding to the energy of the maximum of the reflectivity curves. Reflectance values are determined with an experimental uncertainty of 1%.

## 3. Results

Figure 2 shows the θ–2θ reflectivity curves obtained for six photon energies from ~47 to ~57 eV. Reflectance is quite large, around 0.3. As expected from the Bragg law, the position of the peak shifts toward low angles when the photon energy increases. The increase in reflectivity with increasing radiation energy is associated primarily with a decrease in the absorption of all materials present in the stack.

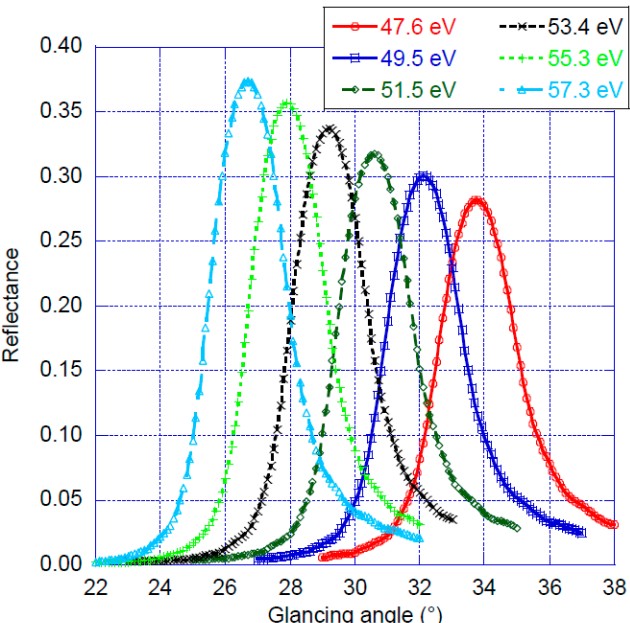

**Figure 2.** Reflectivity curves (in linear scale) of the Be/Si/Al multilayer as a function of the glancing angle for different photon energies spanning the range of the Li K emission.

The reflectivity curves measured as a function of photon energy, from 40 eV up to 195 eV, for glancing angles corresponding to five maxima of angular curves of Figure 2 are shown in Figure 3. The first order diffraction peak is observed around 50 eV, after which the reflected intensity decreases steadily. There are no clear high order diffraction peaks; this confirms the advantage of using the three selected materials. Large variations of the reflectance are observed in the range of the Al L2,3 (around 73 eV), Si L2,3 (around 99 eV) and Be K (around 112 eV) absorption edges, related to drastic changes of the optical indices of the corresponding elements in these photon energy ranges.

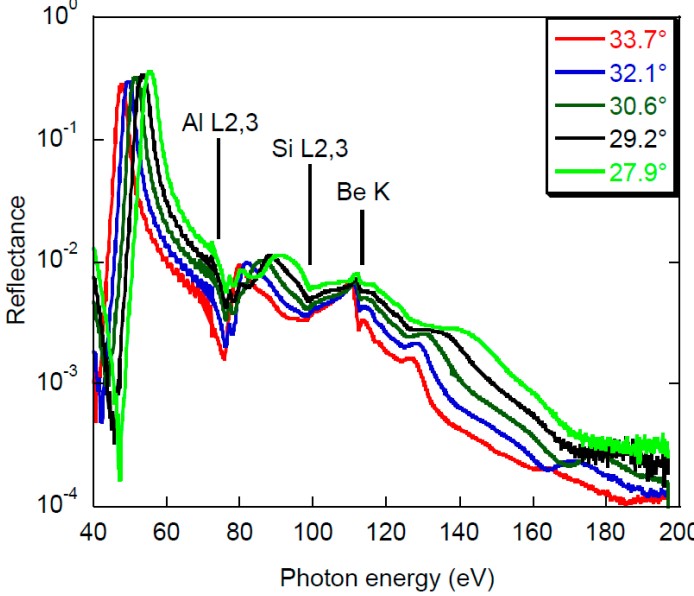

**Figure 3.** Reflectivity curves (in log scale) of the Be-based multilayer as a function of the incident photon energy for different glancing angles corresponding to five maxima of Figure 2. The vertical bars mark the position of the Al, Si and Be absorption edges.

For the five curves of Figure 3, Table 1 gives the bandwidth, determined as the full width at half maximum of the first order diffraction peak, and the rejection rate, determined as the ratio of the reflectance measured at the maximum of the peak (R1) to the reflectance measured at twice (R1/R2) or three times (R1/R3) this energy. The reflectance values are the ones determined at the maximum of the angular reflectivity curves in Figure 2. The rejection of the third order diffracted radiation is about ten times more efficient than the rejection of the second order.

**Table 1.** Parameters of the reflectivity curves shown in Figure 3. Ri is the reflectance measured at the i-th diffraction order. The uncertainty is estimated to be around 0.1 eV for the bandwidth, 1% for R1 and 5% for the reflectance ratios.

| Photon Energy (eV) | Angle (°) | Bandwidth (eV) | R1 | R1/R2 | R1/R3 |
|---|---|---|---|---|---|
| 47.6 | 33.7 | 3.1 | 0.282 | 82 | 737 |
| 49.5 | 32.1 | 3.3 | 0.300 | 83 | 667 |
| 51.5 | 30.6 | 3.5 | 0.318 | 64 | 432 |
| 53.4 | 29.2 | 3.8 | 0.336 | 58 | 618 |
| 55.3 | 27.9 | 4.1 | 0.357 | 50 | 643 |

## 4. Discussion

With respect to the requirements presented in the Introduction regarding the suitability of a periodic multilayer for X-ray spectroscopy in the Li K range, the proposed trilayer stack:

- has a quite high reflectance (0.32 at 51.5 eV, for example); let us note that the measured reflectance is lower than the one expected for a perfect multilayer, i.e., without roughness or interdiffusion (0.404 @ 51.5 eV), so there is still some room to improve the reflectance; an optimized reflectance is mandatory when working in this spectral range, as the K fluorescence yield of lithium is very low, $9 \ 10^{-5}$ [25], and as the windows used in front of the detector or between the sample and spectrometer chambers of EPMA absorb many useful photons; if no care is taken to choose these windows, their transmission can be as low as $10^{-10}$ (a polymer having a density of 1.11 g.cm$^{-3}$, such as polyimide, having a total thickness of 2 µm), jeopardizing the measurement of the Li K emission; this transmission can be as high as $5 \ 10^{-3}$ if the setup is optimized (suppression of the window between the sample and spectrometer chambers, use of a low density polymer of 0.9 g.cm$^{-3}$, such as polypropylene, having a thickness of 0.5 µm and supported by a grid having a 50% transmission);
- presents a quite narrow bandwidth (mean of 3.5 eV, leading to a resolving power $E/\Delta E = 0.07$), see Table 1; this value is smaller than the width of the Li K emission band, which can be 10 eV wide [5–9]; thus, it should be possible to obtain an idea of the chemical state of the lithium atoms from the examination of the shape of the Li K spectrum and its comparison to those of the reference samples, whereas with a grating spectrometer having a resolving power of 0.01 or better, it is possible to determine the chemical state of a lithium atom directly from the examination of the shape of its Li K emission band;
- rejects the radiation diffracted at high orders (see Table 1 and Figure 3) moderately for the second order and efficiently for the third order; the rejection of the second order diffracted radiation with a perfect multilayer is expected from simulations to be between 150 and 200, i.e., two or four times better than the measured one;
- works around a glancing angle around 30°, which is in the middle of the angular range scanned by standard spectrometers equipping EPMA.

The Li K emission corresponds to the electronic transition between the Li *2p* valence states to Li *1s* states. Thus, it is an emission band whose shape (its energy distribution) describes the density of occupied valence states. As a consequence, the natural width of

the emission is closely related to the width of the valence band, i.e., up to 10 eV wide (for example 4 eV in the case of Li metal [26]). The shape of the emission band is sensitive to the chemical state of the lithium atom, as the Li *2p* valence electrons are the ones participating in the chemical bond. This sensitivity is very good when observing the Li band with a grating spectrometer, introducing a spectral broadening smaller than 1 eV. The bandwidth of the designed multilayer is around 3.5 eV. This will lead to a broadening of the emission band. However, because the width of the diffraction pattern of the multilayer is smaller than the width of the valence band, the shifts and variations of shape of the main features of the valence band will be observed, and thus, chemical information about the Li atoms present in the studied sample will be obtained. Moreover, EPMA are mainly devoted to performing quantification for which the important quantity to measure is the intensity, which does not require high spectral resolution.

Let us note that the Li K emission has already been obtained in energy dispersive mode using a silicon drift detector [27]. This allowed the qualitative analysis of a few compounds to be performed. In this case, the poor spectral resolution cannot afford resolving, for example, the Li K and Al L emissions, despite both elements being present in many compounds (aluminosilicates for example). This is one reason why working in wavelength dispersive mode with periodic multilayers is interesting. The performance of the multilayer developed for the spectral range of the Li K emission will lead to improved determination of the shape and intensity of this emission band. Thus, we expect that the identification of the chemical state of lithium atoms and accurate lithium quantification will be performed on SEM and EPMA. This will be a major advance in X-ray microanalysis.

**Author Contributions:** Conceptualization, N.R., E.B. and P.J.; methodology V.P., R.P., A.G. and P.J.; validation, V.P., R.P., A.G. and P.J.; formal analysis, R.P.; investigation, A.G., K.L.G., I.I. and P.J.; data curation, V.P., R.P. and A.G.; writing—original draft preparation, V.P. and P.J.; writing—review and editing, all coauthors; funding acquisition, N.C. and P.J. All authors have read and agreed to the published version of the manuscript.

**Funding:** This research was partially funded by Agence Nationale de la Recherche, project SQLX (ANR-20-CE29-0022) and by state order no. 0030-2021-0022, in part related to the experimental deposition of the Be/Si/Al multilayers. The research leading to this result has been supported by the project CALIPSOplus under Grant Agreement 730872 from the EU Framework Programme for Research and Innovation HORIZON 2020.

**Institutional Review Board Statement:** Not applicable.

**Informed Consent Statement:** Not applicable.

**Acknowledgments:** The soft X-ray experiment was carried out in proposal 20200015 at Elettra.

**Conflicts of Interest:** The authors declare no conflict of interest.

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
