# Peer review of "Periodic Multilayer for X-ray Spectroscopy in the Li K Range"

_applsci, doi:10.3390/app11146385_

Round 1
Reviewer 1 Report
The authors present a multilayer optic suitable for Li edge X-rays. This is potentially interesting, however I feel that more details and discussion/comparison about exactly which situations and exactly how this is superior to grating optics are needed for clarity.
In particular multilayer optics in this energy range are quite difficult because penetration depths are quite shallow and so only very few layers are ever encountered. How many layers are actually used/necessary in this case?
Finally the resolution/bandwidth seems quite large to me, and I would like much more discussion of how this limits applications. In particular it would seem that near edge spectra will be smeared out considerably.
Author Response
The authors present a multilayer optic suitable for Li edge X-rays. This is potentially interesting, however I feel that more details and discussion/comparison about exactly which situations and exactly how this is superior to grating optics are needed for clarity.
In particular multilayer optics in this energy range are quite difficult because penetration depths are quite shallow and so only very few layers are ever encountered. How many layers are actually used/necessary in this case?
The referee is true, penetration depths are shallow owing to a large attenuation lengths. In fact, the number of layers was indicated in the first paragraph of the Materials and Methods section. However, it was written with letters. Now we indicate with digits that there were 20 periods and that this number corresponds to 60 layers. Moreover, we add in the corresponding paragraph that we choose this number of periods because it gives a reflectance very close to the maximum reflectance expected with a larger number of periods.
Finally the resolution/bandwidth seems quite large to me, and I would like much more discussion of how this limits applications. In particular it would seem that near edge spectra will be smeared out considerably.
As stated in the second point of the discussion and in the paragraph following the four points, we indicated that with the obtained bandwidth of the multilayer and taking into account the natural width of the valence band of lithium compounds, it will be possible to get an idea of the chemical state of the emitting lithium atom. We now add that in point 2 that with a grating spectrometer having a resolving power of 0.01 (0.5 eV bandwith) or better it is possible to determine the chemical state of a lithium atom directly from the examination of the shape of its Li K emission band. We also add at the end of the paragraph following the four points that EMPA is mainly used to perform quantification, relying on intensity measurements and which does not require high resolution.
Reviewer 2 Report
The manuscript is well written and the results are shown in clear way.
Some minor changes I suggest:
- there's no information about energy resolution of the multilayer
- some colours of the curves in fig 2 & 3 should be changed since they are either too bright or too similar to distinguish one from another
- it should be pointed out how the maximum of the peak of reflectance was established (the value in Table 1 is given with very high precision)
Author Response
Referee 2
The manuscript is well written and the results are shown in clear way.
Some minor changes I suggest:
- there's no information about energy resolution of the multilayer
In fact, the resolving power is given by the ratio of the bandwidth (which was given in the text) to the photon energy. So, we add in the discussion that we reach a resolving power of 0.07.
- some colours of the curves in fig 2 & 3 should be changed since they are either too bright or too similar to distinguish one from another
We changed the colours, used thicker lines and increased the size of the symbols in figure 2. We changed the colour of one curve in figure 3.
- it should be pointed out how the maximum of the peak of reflectance was established (the value in Table 1 is given with very high precision)
We give the reflectance values with 3 significant digits because as stated in the caption of Table 1, we estimate the uncertainty to about 1%. For sure, as Figure 3 is presented, in log scale, it will be difficult for the reader to extract the indicated reflectance values from the plotted curves. However, during the experiment, the zones where large reflectance variations occur were scanned with an energy step of 0.1 eV, which enabled us to clearly define the maximum of the reflectivity curves and thus determine precisely the reflectance value. Moreover, there is a correspondence between maximum obtained in figure 2 (reflectance at a given energy as a function of the angle) and in figure 3 (reflectance at a given angle as a function of the energy). Thus, the reader can easily extract the reflectance values from figure 2. This is clearly stated in the paragraph discussing Table 1 as “The reflectance values are the ones determined at the maximum of the angular reflectivity curves in Figure 2.”.
Reviewer 3 Report
In the present work, the authors propose the use of an Al/Si/Be multilayer structure for wavelength-dispersive x-ray spectroscopy in the Li K range. The lack of suitable natural crystals in this energy range makes the search for the proper multilayer structure highly desirable. The multilayer development and its characterization demonstrates that except for the elemental detection of lithium, the reported energy resolution may allow the chemical speciation of the Li atoms. The missing part is a Lithium spectrum to demonstrate the full realization of the project. Nevertheless, the subject is of high interest, the paper is well written, and the goal is well-realized. Consequently, I recommend its publication.
As minor points:
a) an error estimation or discussion concerning the thickness layers in lines 107-108,
b) at least a reference is required concerning the Li measurement with a grating spectrometer (lines 180-181),
c) a comment concerning the absence of lithium K transition spectrum. The extremely low fluorescence yield, in combination with the detection efficiency of a crystal spectrometer, background noise, e.t.c, may finally jeopardize the measurement of K lithium transition?
d) the format of the figures must be homogenized (in figure 1, the axes box must be closed, in figure 2, the "degrees" are missing in the x-axis
e) in Table 1, I propose R1/R2 instead of R(n=1)/R(n=2), ...
Author Response
Referee 3
In the present work, the authors propose the use of an Al/Si/Be multilayer structure for wavelength-dispersive x-ray spectroscopy in the Li K range. The lack of suitable natural crystals in this energy range makes the search for the proper multilayer structure highly desirable. The multilayer development and its characterization demonstrates that except for the elemental detection of lithium, the reported energy resolution may allow the chemical speciation of the Li atoms. The missing part is a Lithium spectrum to demonstrate the full realization of the project. Nevertheless, the subject is of high interest, the paper is well written, and the goal is well-realized. Consequently, I recommend its publication.
As minor points:
- a) an error estimation or discussion concerning the thickness layers in lines 107-108,
In the paragraph below figure 1 we now indicate that the uncertainty on thickness and roughness is 0.1 nm.
- b) at least a reference is required concerning the Li measurement with a grating spectrometer (lines 180-181),
In fact, we add the precision in the second paragraph of the Introduction that Refs 5-9 were obtained with grating spectrometers.
- c) a comment concerning the absence of lithium K transition spectrum. The extremely low fluorescence yield, in combination with the detection efficiency of a crystal spectrometer, background noise, e.t.c, may finally jeopardize the measurement of K lithium transition?
It is right that the detection efficiency of a crystal spectrometer is low, coming mainly from the small solid angle of collection of the emitted x-rays. However, there is no problem to use such spectrometers in various spectral ranges to quantify impurities (down to a few ppm). Because the solid angle of the spectrometer equipped with the proposed multilayer will be the same as equipped with crystals or other periodic multilayers, we think this will not the main reason making the measurement of the LiK emission difficult. As mentioned in the first point of the discussion, we think that the low fluorescence yield (on which we cannot play) and the absorption in the windows present inside the spectrometer are the main sources of difficulty. Then, we add a short discussion in this point, explaining that if no special care is taken for the choice of the windows, their transmission can be as low as 10-10, but as high as 5 10-3 with an optimized setup.
- d) the format of the figures must be homogenized (in figure 1, the axes box must be closed, in figure 2, the "degrees" are missing in the x-axis
In Figure 1 the axes are now closed. In figure 2, we use “Glancing angle (°)” as the legend of the x-axis.
- e) in Table 1, I propose R1/R2 instead of R(n=1)/R(n=2), ...
We change “R (n=i)” ro “Ri” in the Table 1. We change the caption accordingly.
Reviewer 4 Report
This work presents an effective technocal solution for a relevant problem in x-ray applications. The approach is reasonable and the results are convincing. I not in particular the clever use of the silicon layer to decrease the roughness the final structure. All practitioners of x-ray techniques will be potentially interested in the results. I recommend acceptance in the present form
Author Response
Referee 4
Comments and Suggestions for Authors
This work presents an effective technocal solution for a relevant problem in x-ray applications. The approach is reasonable and the results are convincing. I not in particular the clever use of the silicon layer to decrease the roughness the final structure. All practitioners of x-ray techniques will be potentially interested in the results. I recommend acceptance in the present form
Thank you !
We take the opportunity of this revision to add a reference (16) about the use of periodic multilayers for spectroscopic applications.